# Potential mechanisms and effects of AFB₁-induced asthma: A comprehensive analysis based on network toxicology and molecular docking

Zhiyue Yu[1,☯], Ming Gao[1,☯], Xiaowen Wu[1], Jiaxiang Li[1], Baijun Liu[1], Rui Wang[1], Jing Tan[1], Xiangyan Yu[2]*, Limei Geng[2]*

**1** Graduate School, Hebei University of Chinese Medicine, Shijiazhuang, Hebei, China, **2** Department of Respiratory Medicine, The Traditional Chinese Medical Hospital of Hebei Province, Shijiazhuang, China

☯ These authors contributed equally to this work.
* glm2018@126.com (LMG); 1308100042@qq.com (XYY)

## Abstract

The aim of this study was to systematically explore the potential molecular mechanisms by which aflatoxin B1 (AFB₁) may trigger asthma using network toxicology and molecular docking. Potential targets related to asthma caused by AFB₁ were obtained from databases, such as PubChem, ProTox, ADMETab, and GeneCards. The targets most significantly related to asthma were further screened using STRING and Cytoscape and analyzed using GO and KEGG enrichment. Finally, molecular docking and visualization were performed using AutodockVina 1.2.2 and PyMOL 2.5 to further determine the affinity between AFB₁ and the core targets. We identified 31 potential targets associated with AFB₁ exposure and asthma, including PTGS2, ADRB2, CysLTR1, PTGS1, and others. The enrichment analysis revealed that the core targets of AFB₁-induced asthma were neuroactive ligand–receptor interactions, the calcium signaling pathway, and the adipocyte catabolism-related signaling pathway. Molecular docking results revealed that AFB₁ exhibited good affinity for the core targets. In the present study, the potential mechanisms involved in AFB₁-induced asthma were elucidated, new insights into how environmental toxins trigger asthma were provided, and a theoretical foundation for asthma prevention and treatment was established.

## 1. Introduction

Food safety issues have always been a major challenge in the field of global public health [1,2,3]. AFB₁, one of the most toxic mycotoxins with the highest risk of carcinogenicity, has become a key focus in the prevention and control of contamination in food safety systems [3]. Studies have shown that AFB₁ is structurally stable, lipophilic, and has significant respiratory adsorptive capacity [4,5]. When combined with hull dust in airborne dust, it easily enters the human body through the respiratory

**Data availability statement:** The datasets generated dring the current study are available in the zenodo respository, https://doi.org/10.5281/zenodo.17758079.

**Funding:** This work was supported by Hebei Provincial Finance Department Project [No. 361-0402-YSN-ULZ6] .The funders had no role in study design, data collection and analysis, decision to publish, or preparation of the manuscript.

**Competing interests:** The authors have declared that no competing interests exist.

system, thus affecting the respiratory system of exposedorganisms [5,6]. $AFB_1$ can act directly on the respiratory system through the P450 metabolic pathway in alveolar cells, inducing respiratory disorders such as sinusitis, noninvasive fungal pneumonia, and other respiratory disorders [7]. Alternatively, $AFB_1$ can also be detected in the respiratory tract through RIG-I and p38-mediated RETREG1/FAM134B-dependent endoplasmic reticulum phagocytosis mechanisms, leading to lung tissue injury [8,9]. Furthermore, $AFB_1$ is directly cytotoxic and inhibits the viability of airway epithelial cells (e.g., A549) and macrophages (THP-1), disrupting cilia function and leading to airway barrier damage [10,11]. Moreover, $AFB_1$ can also induce lung cell DNA damage, binding to DNA via metabolically generated epoxides and increasing the risk of genetic mutations, which is closely related to lung carcinogenesis in occupationally exposed populations [12,13]. Moreover, $AFB_1$ and its extracts can differentially regulate immune responses. Specifically, AFB1 promotes the release of IL-6/IL-8, triggering neutrophil infiltration, and $AFB_1$ extracts upregulate TNF-α/IL-1β/IL-17, causing an inflammatory imbalance, which exacerbates chronic inflammation of the respiratory tract [11,14]. In vivo experiments further confirmed the toxic effects of $AFB_1$ on the lungs and its immunomodulatory effects. Research has shown that exposure to $AFB_1$ significantly increases the level of DNA oxidation and the expression of 8-oxoguanine DNA glycosylase 1 in mouse lung tissue, suggesting that $AFB_1$ promotes lung lesions through oxidative stress and disruption of DNA repair mechanisms [15,16]. Moreover, $AFB_1$ exerts dual effects on immune regulation. Short-term exposure enhances the M1-type alveolar macrophage response in the respiratory system, exacerbating early influenza virus infection and the inflammatory process, whereas long-term exposure may trigger immune suppression [17]. This disruption of immune homeostasis can act synergistically with oxidative stress and inflammatory responses to induce allergic diseases such as allergic pneumonitis and allergic bronchial pulmonary aspergillosis and, in severe cases, can lead to lung malignancy [4,18,19].

Notably, asthma has the highest incidence rate among chronic respiratory diseases (CRDs), accounting for 69% of new CRD cases worldwide [20]. Its symptoms, such as wheezing, shortness of breath, coughing, and chest tightness, are often triggered or exacerbated by exposure to allergens, such as mold, dust mites, pollen, or air pollution [21–27]. As a common allergen, mold has a particularly noteworthy effect on the respiratory system [28]. Studies have shown that fungal spore particles are small and can easily penetrate the lower respiratory tract, triggering severe inflammatory responses, particularly in environments with high fungal concentrations, where the incidence of acute asthma attacks and hospitalization rates increase significantly [29]. Additionally, fungal toxins can disrupt the local immune microenvironment in the lungs, triggering cytokine storms and airway hyperresponsiveness (AHR), and play a significant role in the pathogenesis of asthma [30,31]. Fungi can also directly activate innate immune pathways, particularly the β-glucan–TH17 and chitin–ILC2 pathways, further exacerbating airway inflammation and contributing to the development of steroid-resistant asthma [22]. Among the numerous fungi, $AFB_1$, a mycotoxin subject to the strictest global legislative control, poses a serious threat to human

health. However, systematic research on the effects of this potent mycotoxin in asthma and its toxicological mechanisms is lacking and warrants further exploration [32].

Therefore, the aim of this study was to use network toxicology methods to systematically explore the potential molecular mechanisms by which AFB$_1$ induces asthma. By integrating bioinformatics, molecular docking, and other techniques, we hope to identify the key core targets of AFB$_1$- induced asthma, clarify the important signaling pathways that may be affected, offer new insights into the pathological mechanisms of AFB$_1$-induced asthma, and help develop relevant prevention and treatment strategies.

## 2. Materials and methods

### 2.1. Potential targets of AFB$_1$ and asthma

The 3D structure and physicochemical properties of AFB$_1$ were obtained from the PubChem database [33] (https://pubchem.ncbi.nlm.nih.gov/). Toxicity analysis of AFB$_1$ was performed using the ProTox [34] (https://tox.charite.de/) and ADMETab [35] (https://admetmesh.scbdd.com/) databases, and target prediction for AFB$_1$ was performed using chEMBL [36] (https://www.ebi.ac.uk/chembl/), STITCH [37] (http://stitch.embl.de/), and SwissTargetPrediction [38] (http://www.swisstargetprediction.ch/). The identified targets from these databases were integrated, and duplicates were removed. The remaining targets were batch-converted to a standardized format using the UniProt database [39] (https://www.uniprot.org/).

Potential asthma-related targets were identified using GeneCards (https://www.genecards.org/), and the predicted results were then exported. Targets collected from the GeneCards database were filtered on the basis of a relevance score ≥10 and combined with targets from the OMIM (https://omim.org) and TTD (http://db.idrblab.net/ttd/) databases to construct an asthma-related gene set for further analysis. The target genes of AFB$_1$ were intersected with asthma-related genes, and a Venn diagram was constructed.

### 2.2. Protein-protein interaction (PPI) network

We imported the intersecting genes of potential AFB$_1$-induced asthma targets into the STRING database, with the biological species set to "*Homo sapiens*," and the "minimum interaction score" set to "medium confidence > 0.4", to construct a PPI network that reflected the physical and functional interactions between proteins. Results in the.tsv format were introduced into Cytoscape v3.10.3 to visualize the data and extract the hub genes according to degree.

### 2.3. Enrichment analysis

Gene Ontology (GO) and Kyoto Encyclopedia of Genes and Genomes (KEGG) enrichment analyses were performed to understand gene regulation and function. GO analysis was utilized to screen biological processes (BPs), cellular components (CCs), molecular functions (MFs), and signaling pathways. Additionally, KEGG enrichment analysis was used to identify crucial signaling pathways involved in biological processes. The data obtained from the GO and KEGG pathway enrichment analyses were uploaded to an online data analysis and visualization platform (https://www.bioinformatics.com.cn/) for further processing. Statistical significance was corrected using the Benjamini-Hochberg FDR method, and we considered a P value below 0.05 as meaningful. We also used bubble plots to show the GO and KEGG results.

### 2.4. Molecular docking

To evaluate the binding energies and interaction patterns between the candidate drugs and their respective targets, we employed AutoDock Vina 1.2.2, a computational protein–ligand docking software.

The X-ray crystal structures of ADRB2 (PDB ID: 2R4R), CYSLTR1 (PDB ID: 6RZ4), PTGS1 (PDB ID: 6Y3C), and PTGS2 (PDB ID: 5F19) were obtained from the RCSB Protein Data Bank, while the 3D structure of AFB$_1$ (CAS: 1162-65-8)

was retrieved from the PubChem database. Protein and ligand files were preprocessed by converting all structures into PDBQT format and adding polar hydrogen atoms.

Docking grid parameters are summarized in Table 1, with a grid spacing of 0.05 nm. Molecular docking simulations were performed using AutoDock Vina 1.2.2 with the exhaustiveness parameter set to 32, generating nine binding poses for each ligand–target pair. The binding conformation with the lowest predicted binding energy was selected for subsequent analysis. No positive or negative control compounds were included in this study, as the primary aim was to comparatively evaluate the binding affinities and interaction patterns of the candidate drugs with their respective targets under identical docking conditions. The binding interfaces of the protein–ligand complexes were systematically analyzed using PLIP and LigPlus, and interaction-related details were further examined with PyMOL 2.5.

## 3. Results

### 3.1. Identification of potential targets for $AFB_1$ and asthma

The molecular formula of $AFB_1$ is $C17H12O6$, with six oxygen atoms providing lone pairs of electrons that participate in hydrogen bond formation. Using online databases, we identified potential targets for AFB1 and asthma, finding approximately 188 targets for "$AFB_1$" and approximately 259 targets (Fig 1A) for "asthma." We identified 31 overlapping targets for $AFB_1$ (Fig 1B) and asthma for subsequent analysis.

### 3.2. Construction of a PPI network to identify potential targets of AFB1-induced asthma

The PPI network, which had 31 nodes and 68 edges, was constructed using the intersection of $AFB_1$ and asthma target genes (Fig 2) In this study, PTGS2, ADRB2, MMP9, CysLTR1, and PTGS1 were the top five targets according to degree value, representing the hub targets of $AFB_1$-induced asthma.

### 3.3. GO and KEGG pathway enrichment analysis

The GO and KEGG pathways were visualized and analyzed using an online data platform. We drew several key conclusions from this study. First, we identified several crucial biological processes (BPs) involving the intersecting genes, as shown in Fig 3, such as the regulation of vascular contraction, blood circulation, and G protein-coupled receptors(GPCRs) regulated by adenylate cyclase. The second category entailed cellular components (CCs), such as synaptic membranes, neuronal cell bodies, outer plasma membranes, neuronal projection terminals, dense core vesicles in neurons, and dense core granules. The coexpressed genes presented various molecular functions (MFs), such as GPCR activity, hydrolase activity, adenylate cyclase activity, phospholipid phosphatase C activity, and serine protease activity.

The results of the KEGG pathway enrichment analysis (Fig 4) revealed that $AFB_1$ affected the onset and development of asthma through multiple crucial signaling pathways. These pathways included neuroactive ligand–receptor interactions, calcium signaling, lipolysis regulation in adipocytes, cGMP–PKG signaling, and hormone signaling. Additionally, we sorted the pathways by count values, selected the top four pathways, identified the genes most enriched in these pathways, and

**Table 1. Molecular docking parameters, results and validation of $AFB_1$ with different proteins.**

| Target Protein | Ligand | Box Size (x,y,z) | Center (x,y,z) | Docking Score (kcal/mol) | Redocking Score (kcal/mol) | RMSD (Å) |
|---|---|---|---|---|---|---|
| PTGS2 | $AFB_1$ | 56.0, 62.0, 60.0 | 20.899, 37.499, 59.304 | −8.5 | −8.5 | 0.01 |
| ADRB2 | $AFB_1$ | 40.0, 40.0, 40.0 | 31.902, 52.14, 45.377 | −8 | −8.1 | 0.001 |
| CYSLTR1 | $AFB_1$ | 50.0, 40.0, 106.0 | 33.721, 22.14, 34.482 | −8.7 | −8.9 | 0.001 |
| PTGS1 | $AFB_1$ | 68.0, 54.0, 58.0 | −36.709, −51.733, 2.080 | −7.7 | −7.7 | 1.8 |

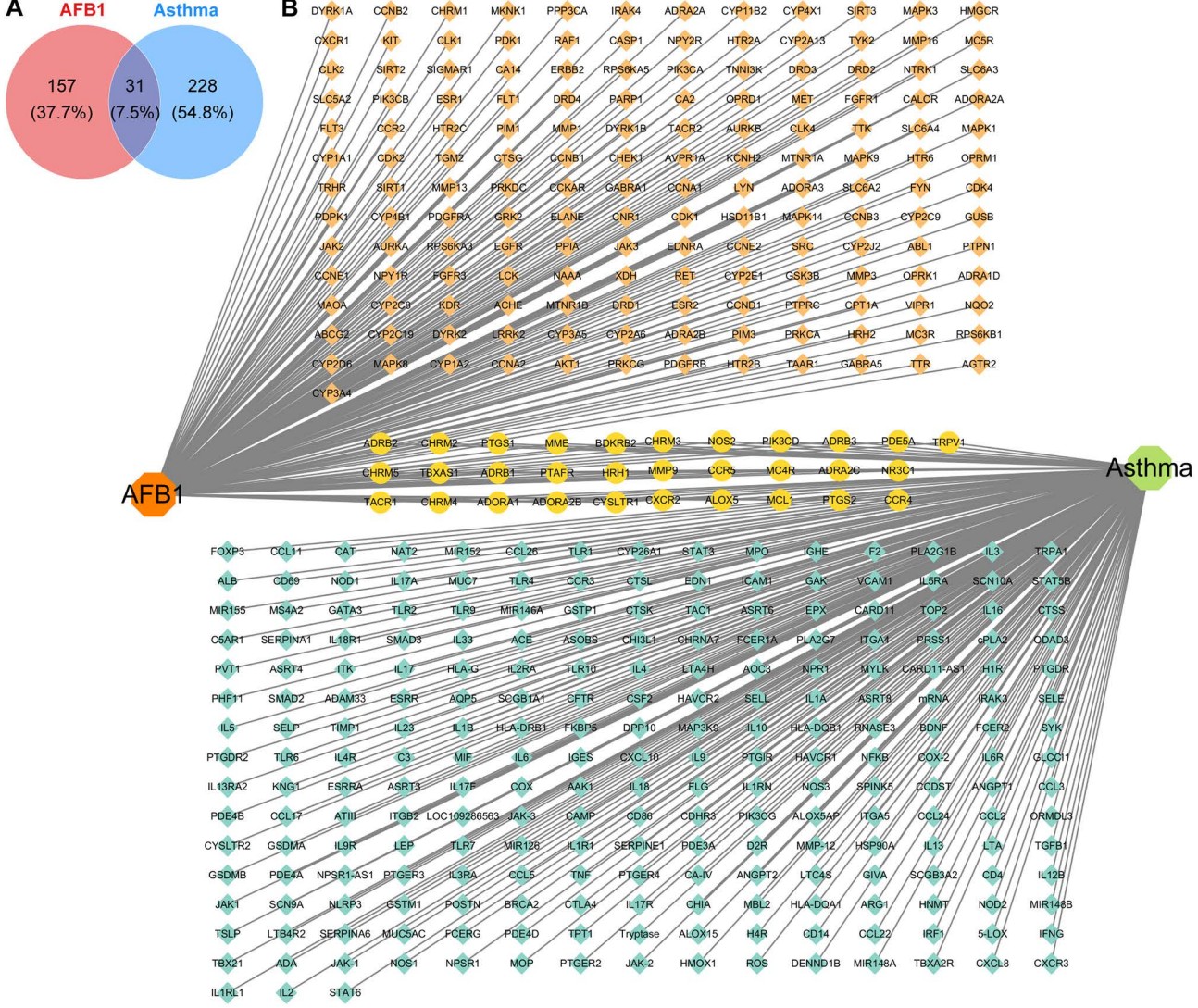

**Fig 1. Analysis of potential overlapping targets between AFB₁ and asthma. (A)** Venn diagram of the targets of AFB₁ and asthma. **(B)** A total of 188 target genes of AFB₁, 259 target genes of asthma, and 31 collective target genes were identified.

intersected them with the PPI core genes to further optimize our PPI network. This process ultimately yielded four core genes, specifically PTGS2, ADRB2, CysLTR1, and PTGS1. We then performed molecular docking between these four genes and AFB1.

### 3.4. Molecular docking

Molecular docking was performed for the PTGS2, ADRB2, CysLTR1, and PTGS1 receptors. The docking results of these targets with AFB₁ are presented in Fig 5 AFB₁ has a strong binding affinity for key targets associated with asthma, characterized by binding energies consistently below the zero threshold (Table 1). This finding not only highlights the potential of AFB₁ to form complexes spontaneously with each of these important target proteins but also suggests its possible involvement in the mechanisms that underlie asthma development.

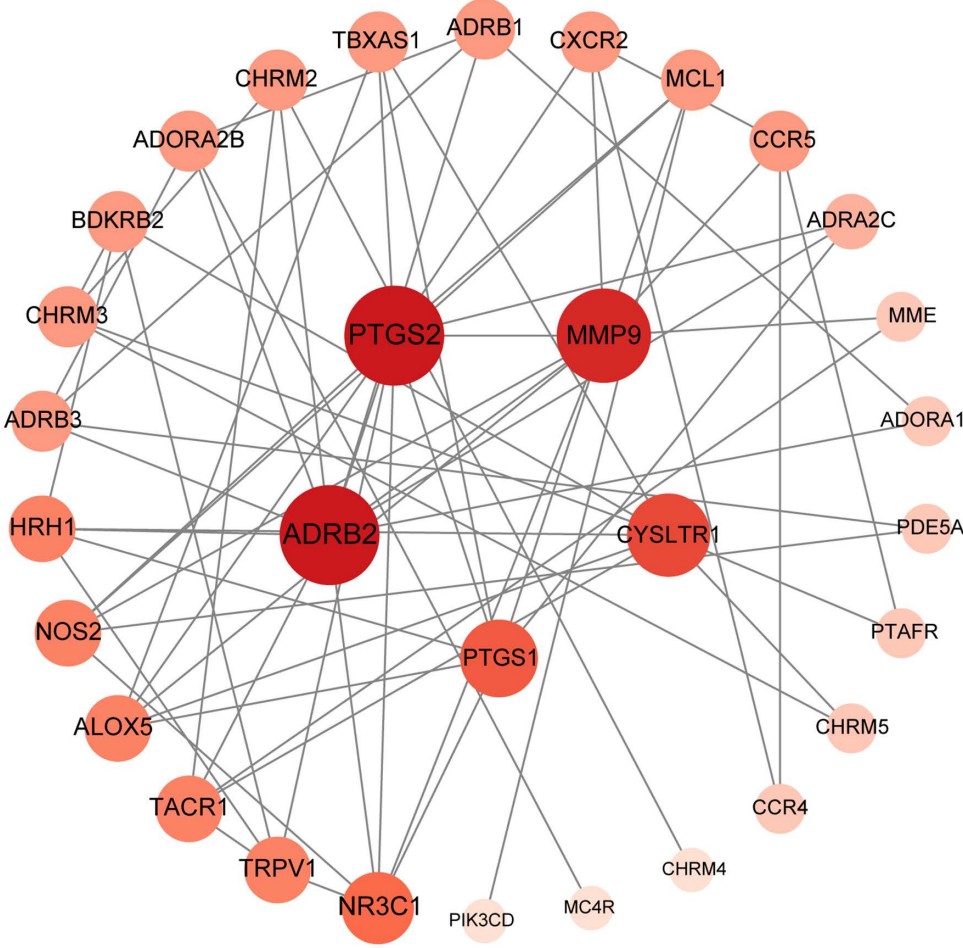

**Fig 2. Collective goal construction of the PPI network, where node size and color represent the node degree values. Larger nodes and darker colors indicate higher node degree values.**

Three-dimensional visual models were constructed for each target protein in the $AFB_1$ small-molecule complex, vividly illustrating the lowest-energy binding conformation and revealing potential hydrogen bond interactions.

## 4. Discussion

Aspergillus is a common allergen that can trigger symptoms in patients with asthma [40]. In this study, we utilized databases such as PubChem, ProTox, ADMETab, and GeneCards to identify 31 potential targets that may be associated with the effects of $AFB_1$ on asthma. Using the STRING platform and Cytoscape software, we constructed a complex interaction network of these potential targets and identified four key nodes: PTGS2, ADRB2, CysLTR1, and PTGS1. These findings help identify potential core targets through which AFB1 may exert its effects on asthma, providing valuable insights into possible mechanisms involved. To predict the binding capacity of AFB1 with its targets, this study employed molecular docking methods to simulate the binding capacity and patterns of $AFB_1$ with the aforementioned key targets. The results indicated that $AFB_1$ may possess potential binding sites corresponding to asthma targets, further providing a reference basis for our network toxicology predictions.

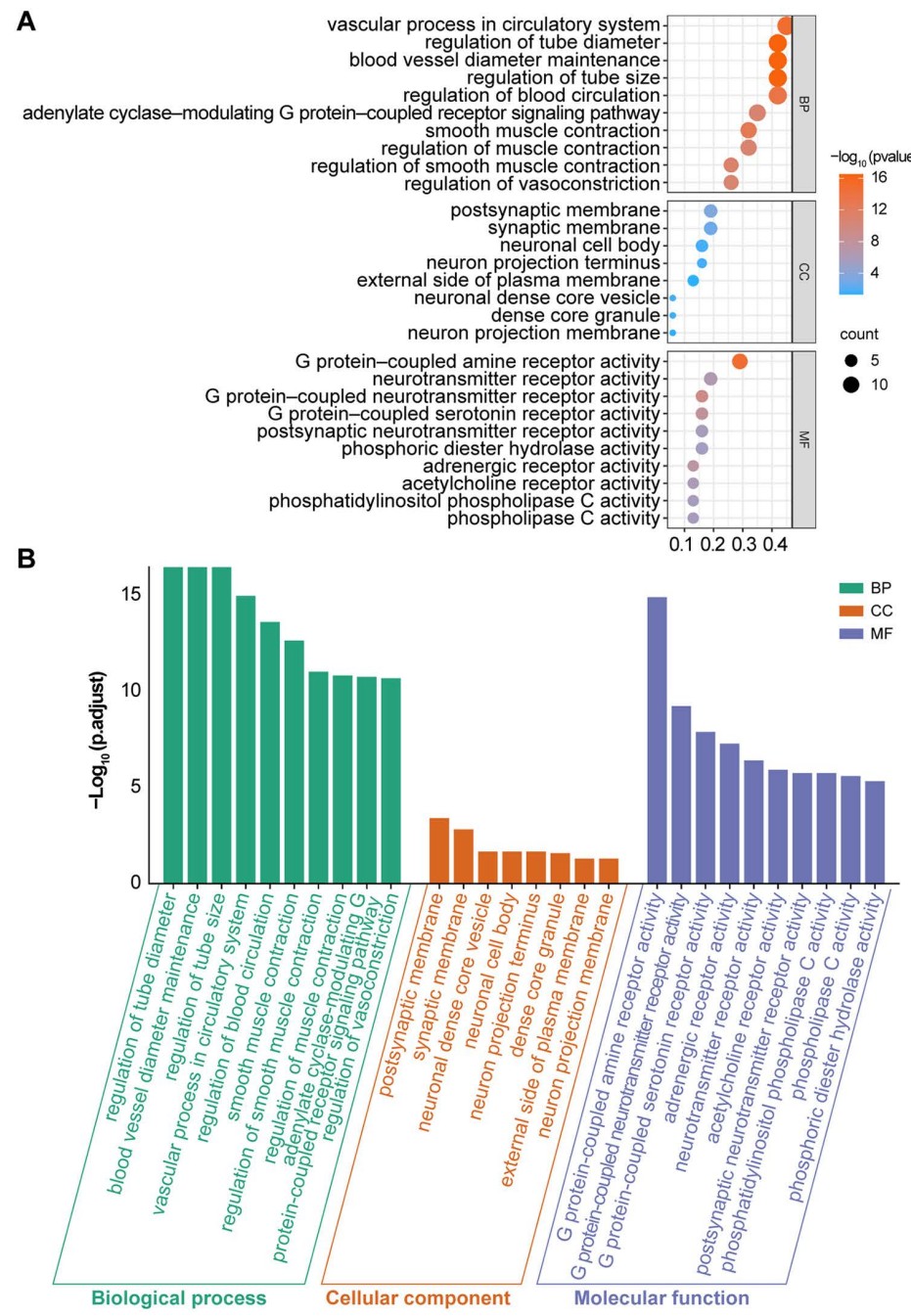

**Fig 3. GO enrichment analysis of potential targets (top 10). (A)** The size of each bubble corresponds to the gene expression in a specific pathway. The color saturation of the bubbles indicates the significance of enrichment. **(B)** Histogram showing the top 10 enriched terms for each GO category (BP, CC, and MF) with small P values across 31 potential targets. P values reflect the statistical significance of enrichment, with lower values indicating greater significance. The height of each bar corresponds to the P value, which reflects the richness within the respective category. These enriched entries highlight the key biological processes, cellular components, and molecular functions potentially affected by AFB$_1$ exposure in asthma.

**A**

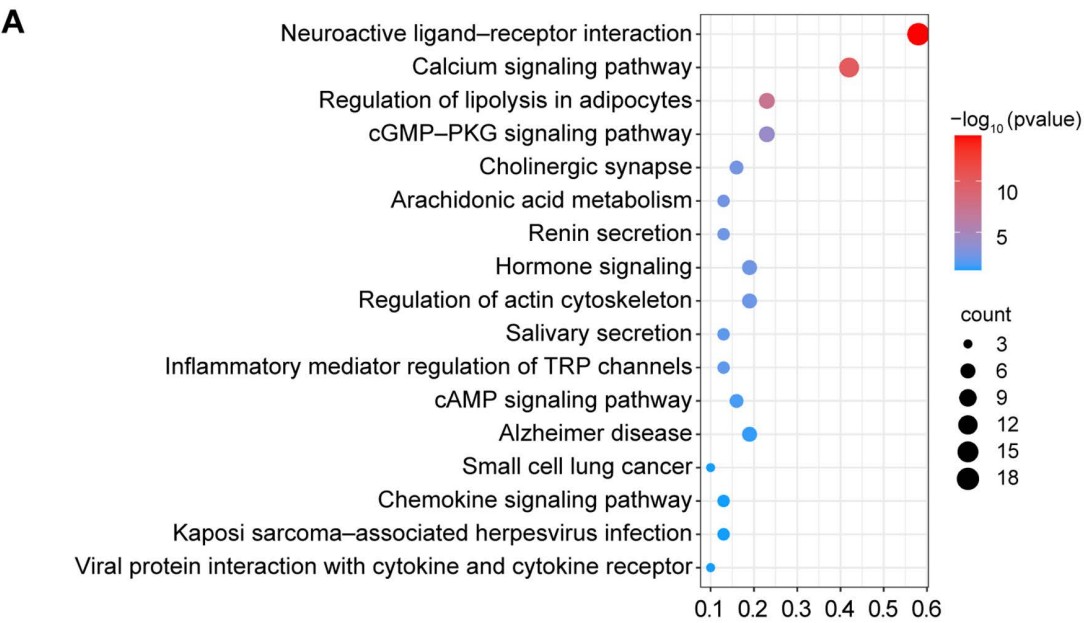

**B**

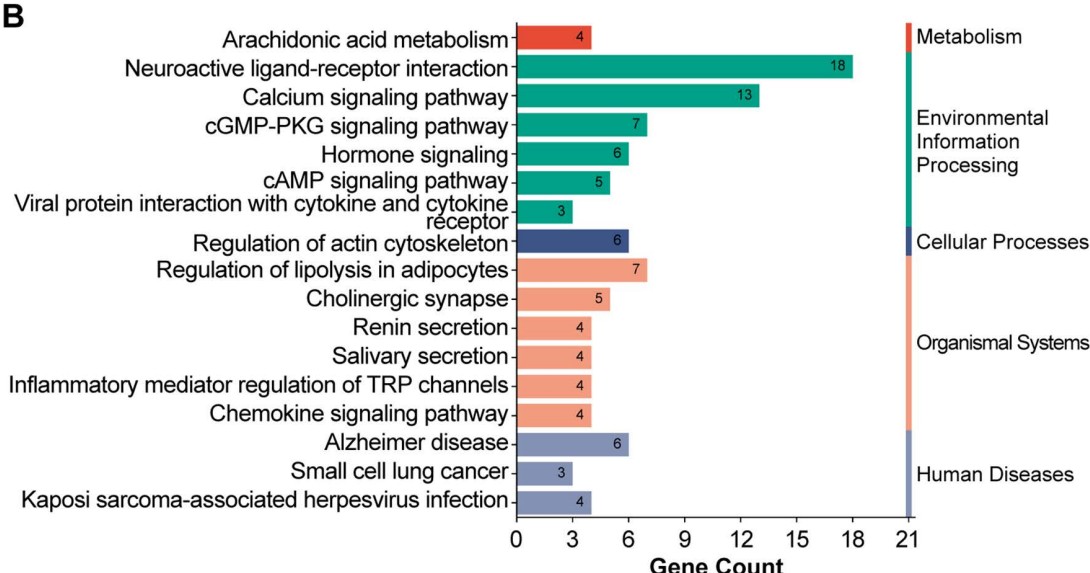

**Fig 4. KEGG enrichment analysis of potential targets. (A)** A bubble chart visualizes the top 20 enriched KEGG signaling pathways in reverse order of the P value. Each bubble represents a specific pathway, and the bubble area indicates the number of enriched genes in that pathway. The intensity of the bubble color indicates the importance of enrichment, with darker red shading indicating greater statistical significance of the pathway. **(B)** Histograms illustrating the enrichment frequency and significance of each pathway. The length of each bar corresponds to the gene count, indicating the enrichment score and significance level, with taller bars representing larger counts and higher enrichment levels.

PTGS2 is a key enzyme in the synthesis of inflammatory prostaglandins (PGs) and catalyzes the conversion of arachidonic acid (AA) into PG, prostacyclin, and thromboxane, thereby participating in inflammatory responses and other physiological processes [41]. PTGS2 plays a significant role in asthma. Clinical studies have shown that, compared with healthy subjects, asthma patients have significantly elevated PTGS2 levels in their bronchoalveolar lavage fluid, suggesting it could be an important biomarker of asthma inflammation activation [42,43]. Additionally, susceptibility to asthma

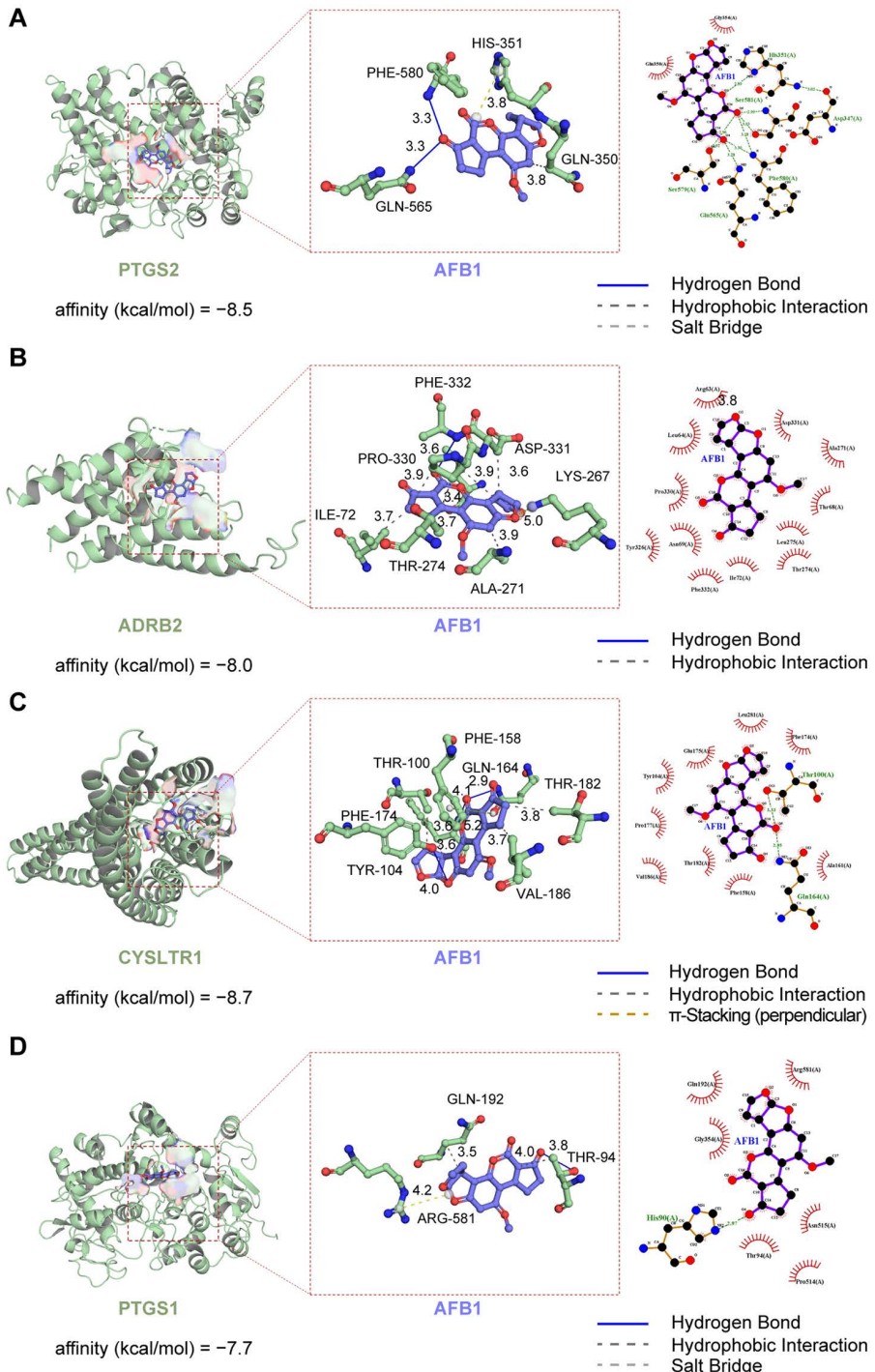

**Fig 5. Molecular docking of each target protein with AFB₁.** (A) AFB₁ and PTGS2, (B) AFB₁ and ADRB2, (C) AFB₁ and CysLTR1, and (D) AFB₁ and PTGS1 (from left to right: 3D and 2D structures).

is closely associated with PTGS2 genetic polymorphisms, for example, a study by Meng, S et al. found that the PTGS2 rs20417 SNP site, which may participate in asthma pathogenesis by binding to transcription factors and influencing the regulation of gene expression [44,45]. Qiu Q et al. identified that PTGS2 can be induced by cytokines, such as IL-1β and TNF-α, amplifying the inflammatory response in asthma. Among these, IL-1β can trigger PTGS2 to induce MUC5AC mucin expression through ERK or p38 MAPK, exacerbating airway obstruction, whereas TNF-α may induce PTGS2 to produce prostaglandin E2 (PGE2) and other substances, exacerbating airway inflammation [46–48]. Additionally, PGE2 produced by PTGS2 can cause desensitization of β2-adrenergic receptors on airway smooth muscle (ASM) cells, impairing airway dilation capacity and exacerbating asthma symptoms [49]. Other studies have shown that PTGS2 is a downstream target of NF-κB and that NF-κB phosphorylation upregulates PTGS2 expression in airway epithelial cells and smooth muscle cells, thereby promoting the release of inflammatory mediators, making it a key step in the pathogenesis of asthma. Moreover, the published studies by Xian Chen et al. followed that environmental exposure may up-regulate the expression of PTGS2 by inducing ferrot in airway epithelial cells [50]. The in vivo experiments by Li et al. provided relatively reliable evidence for this, further showing that in asthmatic mice with elevated $Fe^{2+}$ levels, PTGS2 expression was significantly increased [46]. Ferroptosis can directly increase PTGS2 expression and promote the release of inflammatory signaling molecules, which may contribute to the amplification of inflammatory responses in asthma [51].

ADRB2 is a member of the GPCR family and is highly expressed in bronchial smooth muscle cells. It mediates the physiological effects of catecholamine neurotransmitters in the sympathetic nervous system, thereby relaxing bronchial smooth muscle and inhibiting the release of histamine mediators [52,53]. In respiratory diseases, ADRB2 reduces microvascular leakage and inhibits the release of mediators by certain inflammatory cells, such as neutrophils, eosinophils, and lymphocytes [54,55]. Additionally, the study by Ağaç, D et al. further revealed ADRB2 may directly inhibit TLR-mediated NF-κB activation, thereby suppressing the secretion of TNF-α and IL-12 by innate immune cells and alleviating airway inflammation [56]. Moreover, Lp, N et al. have indicated that in smooth muscle cells, ADRB2 primarily exerts its bronchodilator effects through the Gs-cAMP pathway, whereas in epithelial cells, ADRB2 primarily promotes inflammation and mucus production through the βarr-2 signaling pathway [57]. More importantly, the ADRB2 gene may alter the response of patients with asthma to treatment. In vitro studies have shown that the nonsynonymous single-nucleotide polymorphism at position 46 of the ADRB2 gene (ADRB2 Gly16Arg) significantly enhances the receptor downregulation effect mediated by agonists and may influence the therapeutic efficacy of inhaled corticosteroids, making it an important potential biomarker for personalized asthma treatment [58,59]. Additionally, Fu, A et al.have shown that ADRB2 methylation may reduce ASM responsiveness to beta-receptor agonists by decreasing ADRB2 gene expression, which could be associated with exacerbated asthma symptoms. Furthermore, ADRB2 methylation may increase the sensitivity of patients with asthma to environmental stimuli, thereby increasing the risk of asthma exacerbation [60].

CysLTR1 is a key receptor involved in respiratory system inflammation and is produced by AA, which can stimulate AHR, leading to ASM contraction, increased vascular permeability, and increased mucosal inflammatory secretions [61–64]. Trinh, H et al. found that multiple inflammation-related cells associated with asthma express CysLTR1, and its expression levels are closely correlated with the severity of respiratory tract inflammation [65,66]. Clinical research has confirmed that patients with asthma have significantly higher levels of CysLTR1 miRNA expression and protein-positive cell counts than healthy individuals do. In an inflammatory environment, the upregulation of CysLTR1 expression may increase monocyte sensitivity to cysteinyl leukotrienes (CysLTs), leading to the production of the potent chemokine CCL2, which promotes the aggregation of inflammatory cells in the airways and exacerbates asthma pathophysiology [67]. Further studies by M,P et al. revealed that the CysLTR1 receptor may increase the expression of cell adhesion molecule-1 by activating the phosphorylation of ERK1/2 and STAT-1, thereby promoting the adhesion of eosinophils and playing an important role in chronic inflammation in asthma [68]. Additionally, Parmentier, C. N found that the CysLTR1 receptor can be activated by leukotriene D4, which further triggers the transcriptional activation of the epidermal growth factor receptor through PI3K and ROS, thereby activating the downstream Src-Ras-ERK1/2 signaling pathway and ultimately leading

to the proliferation of ASM cells [69]. CysLTR1 can also bind to CysLTs, leading to increased calcium ion concentrations within Th2 cells, promoting their migration to inflammatory sites, and exacerbating the inflammatory response in asthma [70]. Additionally, studies by Jin,Z.et al. have shown that the CysLTR1 gene (such as the rs320995C allele) can upregulate receptor protein expression, enhancing the therapeutic response to leukotriene modulators and providing a molecular basis for personalized treatment [71].

PTGS1 is a physiologically significant PG synthase that plays a key role in the catalytic conversion of AA to PG and is widely involved in inflammatory responses and the regulation of tissue homeostasis [72–74]. McErlean, P. et al identified that the PTGS1 gene is located within an asthma-associated superenhancer and that its expression is subject to epigenetic regulation [75]. PG synthesis can exacerbate airway inflammation and hyperresponsiveness, thereby promoting asthma development. Clinical studies have revealed that certain transcriptional variants of PTGS1 are significantly upregulated in patients with asthma and that their expression levels are correlated with disease severity [76]. Further research by Allakhverdi,Z.et al. has indicated that in asthma, PTGS1 not only mediates the production of prostaglandins (PGD2, TXA2, LTC4) but also participates in regulating mast cell responses to the type 2 innate immune factor IL-33 [77–79]. Moreover, PTGS1 may be activated by IL-33 to catalyze the production of PG, thereby activating the ERK signaling pathway, such as serine phosphorylation of phospholipase A2, initiating AA release and prostaglandin synthesis, and promoting the development of asthma [80]. However, Gollapudi, R have shown that PGE2 generated by PTGS1 may inhibit the activity of 5-lipoxygenase-activating protein and 5-lipoxygenase, thereby controlling bronchoconstriction, excessive mucus secretion, and eosinophil chemotaxis caused by AA metabolites, such as leukotriene B4, potentially alleviating asthma symptoms to some extent [81]. Additionally, the in vivo experiments by Harrington, L. S. et al. revealed that compared with wild-type mice, PTGS1 gene knockout mice presented significantly elevated lung inflammation markers and enhanced airway responsiveness to bronchoconstrictors, indicating that PTGS1 may inhibit bronchoconstriction and potentially exert a protective role in asthma [82,83].

In this study, the potential functions of these four hub genes were also investigated, and KEGG enrichment analysis revealed that they are involved in neuroactive ligand–receptor interactions, calcium signaling pathways, lipolysis in adipocytes, and cGMP–PKG signaling pathways. Among these, neuroactive ligand–receptor interactions are closely associated with the heterogeneous inflammatory response in asthma, representing a collective set of receptors and ligands on the plasma membrane involved in intracellular and extracellular signaling pathways [84]. For example, neurotransmitters such as acetylcholine and histamine, as well as neuropeptides such as substance P and nerve growth factor, bind to ASM cell receptors, leading to airway constriction, inflammation, and ASM cell proliferation [85]. Moreover, cytokines drive inflammatory cell recruitment through receptor signaling, thereby promoting airway remodeling [86]. Furthermore, abnormal activation of the calcium signaling pathway may significantly influence the pathogenesis of asthma. Studies have shown that such abnormal activation leads to intracellular $Ca^{2+}$ overload and the activation of inflammatory pathways, such as the NF-κB and MAPK pathways, thereby promoting inflammatory responses [87]. Additionally, $Ca^{2+}$ and ROS may exhibit a positive feedback mechanism that further amplifies inflammatory responses [88]. Other studies have shown that lipolysis may be associated with the regulation of inflammation in asthma. Research by Rastogi, D et al. indicates that free fatty acids produced by lipolysis may recruit immune cells and cause a leptin/adiponectin imbalance, triggering chronic low-grade inflammation and exacerbating AHR [89–91]. The cGMP–PKG signaling pathway also plays a crucial role in asthma development and progression. This pathway may promote the opening of large-conductance calcium-activated potassium channels by activating protein kinase-dependent mechanisms, inducing smooth muscle relaxation, and alleviating AHR. Therefore, dysfunction of this pathway may drive airway constriction [92]. These studies suggest that $AFB_1$ may be involved in airway inflammation and remodeling in asthma through neuroactive ligand–receptor interactions, calcium signaling pathways, adipocyte lipolysis, and the cGMP–PKG signaling pathway, potentially contributing to asthma exacerbation.

## 5. Conclusion

In conclusion, in this study network toxicology and molecular docking were used to assess the potential role of $AFB_1$ in asthma. We identified 31 potential targets that may be associated with $AFB_1$ exposure and asthma, including PTGS2, ADRB2, CysLTR1, PTGS1, and others, which play crucial roles in airway inflammation, smooth muscle contraction, immune regulation, and inflammatory mediator production. KEGG analyses suggested that $AFB_1$ may promote airway inflammation and remodeling and even influence drug responses by regulating neuroactive ligand–receptor interactions, calcium signaling pathways, adipocyte lipolysis, and cGMP–PKG signaling pathways. Molecular docking results suggested that $AFB_1$ exhibited good binding activity with the core targets, providing a structural hypothesis for its potential role in asthma. However, this study primarily relies on network analysis and molecular docking simulations, lacking verification through in vivo and in vitro experiments. Future studies should further validate the expression of targets and dynamic changes in signaling pathways through in vivo and in vitro research to clarify the pathogenic mechanisms of $AFB_1$ in asthma.

## Supporting information

**S1 File. Supplementary data tables.** This Excel file contains the following data sheets: 1) Quantitative results of molecular docking, 2) KEGG pathway enrichment analysis, and 3) Top 10 hub targets from the PPI network ranked by degree value.
(XLSX)

**S2 File. Supplementary methods and details.** This document provides detailed information on the versions of all databases and software used, along with extended methodological details and parameters for the molecular docking simulations. The datasets generated dring the current study are available in the zenodo respository, https://doi.org/10.5281/zenodo.17758079.
(DOCX)

## Author contributions

**Conceptualization:** Zhiyue Yu.

**Data curation:** Zhiyue Yu, Ming Gao, Baijun Liu, Rui Wang, Jing Tan.

**Funding acquisition:** Limei Geng.

**Project administration:** Xiangyan Yu, Limei Geng.

**Resources:** Limei Geng.

**Software:** Zhiyue Yu, Ming Gao.

**Supervision:** Limei Geng.

**Visualization:** Zhiyue Yu, Ming Gao, Jiaxiang Li.

**Writing – original draft:** Zhiyue Yu, Xiaowen Wu.

**Writing – review & editing:** Xiangyan Yu, Ming Gao, Xiangyan Yu, Limei Geng.

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
