## [Decision Letter · Decision Letter 0]

11 Nov 2025

Dear Dr. Geng,

Thank you for submitting your manuscript to PLOS ONE. After careful consideration, we feel that it has merit but does not fully meet PLOS ONE’s publication criteria as it currently stands. Therefore, we invite you to submit a revised version of the manuscript that addresses the points raised during the review process.

We look forward to receiving your revised manuscript.

Kind regards,

Fazul Nabi

Academic Editor

PLOS ONE

Journal Requirements:

[Hebei Provincial Finance Department Project].

5. Please upload a new copy of Figures 5A, 5B, and 5C as the details are not clear. Please follow the link for more information: https://journals.plos.org/plosone/s/figures

6. Please include a copy of Table 1 which you refer to in your text on page 9.

7. Please include captions for your Supporting Information files at the end of your manuscript, and update any in-text citations to match accordingly. Please see our Supporting Information guidelines for more information: http://journals.plos.org/plosone/s/supporting-information .

8. We are unable to open your Supporting Information file [AFB1.zip]. Please kindly revise as necessary and re-upload.

Reviewers' comments:

Reviewer's Responses to Questions

**Comments to the Author**

1. Is the manuscript technically sound, and do the data support the conclusions?

Reviewer #1: Yes

Reviewer #2: Yes

2. Has the statistical analysis been performed appropriately and rigorously?

Reviewer #1: Yes

Reviewer #2: N/A

3. Have the authors made all data underlying the findings in their manuscript fully available?

Reviewer #1: Yes

Reviewer #2: No

4. Is the manuscript presented in an intelligible fashion and written in standard English?

Reviewer #1: No

Reviewer #2: Yes

Reviewer #1: Reviewer comments regarding the manuscript titled “Potential mechanisms and effects of AFB1-induced asthma: a comprehensive analysis based on network toxicology and molecular docking”

Title: Seems appropriate to the contents of the manuscript.

Abstract: The abstract is well written with few minor grammatical errors which needs to be addressed before final submission.

Introduction: The manuscript is not having any line numbers so its hard to communicate with the author but there are few grammatical errors such as words with no spacing and wrong grammar in this section. Please address the concern before final submission.

Materials and Methods: This section is clearly written but minor grammar errors needs to be addressed before final submission please.

Results: Please check for grammar and remove English language errors. Please also follow superscript or under script in formula of AFB1. Lines are not justified throughout the manuscript but specifically in this section.

Discussion: This section is well written and explained but please check for grammatical errors.

Conclusion: well written and explained.

No acknowledgment or funding or conflict of interest statement found in the manuscript?

References: Please format references according to the journal requirements.

Figures: would you be able to please improve the quality of figures shown in the manuscript and improve its visibility?

Reviewer #2: General assessment: The manuscript entitled “Potential mechanisms and effects of AFB1-induced asthma: a comprehensive analysis based on network toxicology and molecular docking” explores a novel and under-investigated topic, the potential molecular mechanisms linking Aflatoxin B1 exposure with asthma pathogenesis, using an integrative in silico workflow combining network toxicology, enrichment analysis, and molecular docking. The study is timely, conceptually interesting, and provides biologically plausible hypotheses regarding GPCR signaling, arachidonic acid metabolism, and oxidative stress in AFB1-related airway dysfunction. The manuscript is generally well organized and clearly written. However, several methodological and reporting issues need to be addressed to ensure reproducibility and alignment with PLOS ONE transparency standards.

Major comments:

1. Database and software versioning: please provide detailed versions and access dates for all databases and tools used: ChEMBL, STITCH, SwissTargetPrediction, GeneCards, OMIM, TTD, STRING, ProTox, ADMETlab, and the specific software (R and packages, Cytoscape, AutoDock or equivalent).

2. GO/KEGG enrichment statistics: clearly state the multiple-testing correction method (e.g., Benjamini–Hochberg FDR), the p.adjust threshold used for significance, and attach full result tables (GO/KEGG term ID, description, geneRatio, p.adjust, and gene list) as supplementary CSV files.

3. STRING and PPI network parameters: Specify the confidence score cutoff, the evidence channels included (e.g., experiment, database, text mining), and the organism setting used. Please include the full STRING export (.tsv) as a supplementary file.

4. Hub gene identification: Indicate the algorithm and metric used in CytoHubba (e.g., Degree, MCC, Betweenness) and provide a table listing the top 10 genes with their centrality scores.

5. Molecular docking reproducibility: provide complete details on receptor and ligand preparation, grid box coordinates and size, exhaustiveness, number of poses, and validation steps (e.g., re-docking RMSD < 2 Å). Include a summary table with docking energy values (mean ± SD, kcal/mol), key interacting residues, and hydrogen/hydrophobic bonds. Add positive and negative controls if available.

6. Data and code availability: deposit all raw data, scripts (e.g., Toxicity.venn.R), intermediate files (TXT, TSV, CSV), and figure sources in a public repository such as Zenodo or OSF and update the Data Availability Statement with a DOI link.

7. Discussion and limitations: the Discussion should go beyond result description. Please deepen biological interpretation, include literature comparisons, and explicitly acknowledge the in silico limitations and potential database biases. Rephrase causal statements into hypothesis-generating language (e.g., “may contribute,” “suggests”).

Minor comments

1. Define all abbreviations at first mention (e.g., GPCR, ADMET).

2. Revise the writing for conciseness and grammar consistency; unify past tense throughout Methods and Results.

3. Improve figure legends to include sample sizes, p.adjust scale, and database sources/versions.

4. Include a table summarizing all databases and software (name, version, access date, URL).

5. Cite the official reference articles for the databases and tools (SwissTargetPrediction, STRING, ProTox-II, ADMETlab).

6. Ensure that reference formatting follows PLOS ONE style (numbered citations in square brackets).

The manuscript presents a coherent and potentially impactful in silico investigation. Once the methodological transparency and docking reproducibility issues are resolved, the study would meet PLOS ONE publication standards.

**Do you want your identity to be public for this peer review?** For information about this choice, including consent withdrawal, please see our Privacy Policy

Reviewer #1: **Yes:**  Syed Zahid Ali Shah

Reviewer #2: No

---

## [Author Response · Author response to Decision Letter 1]

1 Dec 2025

Major comments:

Comments1: Database and software versioning: please provide detailed versions and access dates for all databases and tools used: ChEMBL, STITCH, SwissTargetPrediction, GeneCards, OMIM, TTD, STRING, ProTox, ADMETlab, and the specific software (R and packages, Cytoscape, AutoDock or equivalent).

Response1:

Thank you for your valuable comments. We have thoroughly checked and supplemented the version information and access dates for all databases and software used in our study, as requested.

Please find the specific explanations below:

(1)Version Information: We have compiled the complete version details, access dates, and sources into a table in the Supplementary Materials/Methods section. For certain online databases and web tools that do not provide explicit version numbers (e.g., v1.2.3), we adopted the following alternative approaches to ensure traceability:

For the OMIM database, we provided the copyright information displayed on its official page: "Copyright (c) 1966-2025 Johns Hopkins University."

For the TTD database, we provided its most recent relevant citation: "Nucleic Acids Research. 52(D1): 1465-1477 (2024)", to indicate the data version it relies on.

For tools that have been updated but whose core algorithms remain unchanged (e.g., SwissTargetPrediction), we cited their official statement: "...we have NOT changed the underlying technologies and parameters... provides exactly the same results as the previous version", to ensure the reproducibility of the results.

(2)Access Dates: The specific access dates for all online resources have been clearly listed in the table. This provides a precise timestamp for the state of the data at the time of retrieval.

We believe that providing official statements, citation literature, and precise access dates is sufficient to ensure the transparency of our methodology and the reproducibility of the results.

Thank you again for your careful review and constructive feedback.

The table is as follows.

Comments2: GO/KEGG enrichment statistics: clearly state the multiple-testing correction method (e.g., Benjamini–Hochberg FDR), the p.adjust threshold used for significance, and attach full result tables (GO/KEGG term ID, description, geneRatio, p.adjust, and gene list) as supplementary CSV files.

Response2:

Dear Reviewer, thank you for your comments. Regarding the statistical and visualization methods, we have added a corresponding description in the revised manuscript. Please refer to lines 111-113 (highlighted in yellow) in the document "Revised Manuscript with Track Changes". The added text is as follows:

“Statistical significance was corrected using the Benjamini-Hochberg FDR method, and we considered a P value below 0.05 as meaningful. We also used bubble plots to show the GO and KEGG results.”

Additionally, the relevant source data file has been deposited in the Zenodo open repository and is accessible via the following link and DOI: https://doi.org/10.5281/zenodo.17758079

Comments3: STRING and PPI network parameters: Specify the confidence score cutoff, the evidence channels included (e.g., experiment, database, text mining), and the organism setting used. Please include the full STRING export (.tsv) as a supplementary file.

Response3:

Thank you for raising this point. We would like to clarify that the key STRING parameters, including the organism and the minimum interaction score, were already specified in lines 100-101 of the Methods section in our original manuscript.

For your convenience, we summarize the full set of parameters used here:

Organism: Homo sapiens.

Active Interaction Sources: Textmining, Experiments, Databases, Co‑expression, Neighborhood, Gene Fusion, Co‑occurrence.

Minimum Required Interaction Score: 0.400 (Medium confidence)

Query Proteins: TACR1, MC4R, PDE5A, CHRM5, NR3C1, HRH1, CHRM2, ADRB1, BDKRB2, ADRB2, CXCR2, ADRB3, CYSLTR1, ADRA2C, CHRM4, CHRM3, PTAFR, ADORA1, PTGS2, TBXAS1, MMP9, CCR4, CCR5, PTGS1, MME, PIK3CD, ALOX5, ADORA2B, NOS2, MCL1, TRPV1.

Furthermore, as requested, the full STRING export in .tsv format has been deposited as supplementary data in the Zenodo open repository and is accessible via: https://doi.org/10.5281/zenodo.17758079

Comments4: Hub gene identification: Indicate the algorithm and metric used in CytoHubba (e.g., Degree, MCC, Betweenness) and provide a table listing the top 10 genes with their centrality scores.

Response4:

Thank you for your comment. We identified the hub genes using the Degree method in CytoHubba. The top 10 genes and their scores are listed in the table below.

Top 10 in network string_interactions_short.tsv ranked by Degree method

Additionally, this table has been uploaded as a separate supplementary file (filename: degree_top_10) to the system for your review.

Comments5: Molecular docking reproducibility: provide complete details on receptor and ligand preparation, grid box coordinates and size, exhaustiveness, number of poses, and validation steps (e.g., re-docking RMSD < 2 Å). Include a summary table with docking energy values (mean ± SD, kcal/mol), key interacting residues, and hydrogen/hydrophobic bonds. Add positive and negative controls if available.

Response5:

In response to your suggestions, we have further refined the molecular docking section and optimized the corresponding figures. As the content is quite detailed, to maintain the conciseness and readability of the main text, we have moved the entire section to the zenodo under the title “docking” for your review.

Additionally, corresponding revisions have been made in the main text:

Abstract (lines 17–19);

Methods section (2.4, lines 115–127);

Results section (3.4, lines 186–194).

All changes have been highlighted in yellow for your convenience.

We appreciate your guidance and support.

Comments6: Data and code availability: deposit all raw data, scripts (e.g., Toxicity.venn.R), intermediate files (TXT, TSV, CSV), and figure sources in a public repository such as Zenodo or OSF and update the Data Availability Statement with a DOI link.

Response6:

We thank the reviewer for the suggestion. As requested, we have added the 'Data Availability' statement to the main text (Lines 344-346). The added sentence, which reads "The datasets generated during the current study are available in the Zenodo repository, https://doi.org/10.5281/zenodo.17744835" has been highlighted in yellow for your convenience.

Comments7: Discussion and limitations: the Discussion should go beyond result description. Please deepen biological interpretation, include literature comparisons, and explicitly acknowledge the in silico limitations and potential database biases. Rephrase causal statements into hypothesis-generating language (e.g., “may contribute,” “suggests”).

Response7:

We sincerely thank the reviewer for this insightful suggestion. We have revised the Discussion section to deepen the biological interpretation, incorporate comparisons with existing literature, and explicitly acknowledge the limitations of our in silico approach and potential database biases. As advised, causal statements have been rephrased into hypothesis-generating language. All changes have been highlighted in yellow in the revised manuscript. We hope these revisions meet the journal's standards, and we welcome any further suggestions for improvement.

Minor comments:

Comments1: Define all abbreviations at first mention (e.g., GPCR, ADMET).

Response1:

We thank the reviewer for this valuable suggestion. We have systematically checked the manuscript and ensured that all abbreviations are defined at first mention. The corresponding revisions have been highlighted in yellow in the current version at the following lines: 69, 152, 156, 239, 260, 319, and 322. Should any further adjustments be needed, we would be pleased to make them accordingly.

Comments2: Revise the writing for conciseness and grammar consistency; unify past tense throughout Methods and Results.

Response2:

We thank the reviewer for this valuable suggestion. We have carefully proofread the manuscript to improve grammatical consistency and have unified the verb tense to the past tense throughout the Methods and Results sections. All revised areas have been highlighted in yellow for easy identification. We are committed to meeting the journal's standards and would be pleased to make any additional adjustments based on your further guidance.

Comments3: Improve figure legends to include sample sizes, p.adjust scale, and database sources/versions.

Response3:

We sincerely thank the reviewer for their valuable suggestions. We have comprehensively enhanced the content of all figure legends and made formatting adjustments to align with submission requirements. The revised figure files have been deposited in the Zenodo repository (https://doi.org/10.5281/zenodo.17744835) for your review. Detailed version information for the databases and tools used is available in the Supplementary Material under the section “Database and software versioning.” We trust that these revisions have fully addressed the comment raised and welcome any further feedback.

Comments4: Include a table summarizing all databases and software (name, version, access date, URL).

Response4:

We sincerely thank the reviewer for this valuable suggestion. As requested, detailed version information for all databases and tools used in this study has been compiled and is now available in the Supplementary Material under the section titled "Database and Software Versioning" for your review.

Comments5: Cite the official reference articles for the databases and tools (SwissTargetPrediction, STRING, ProTox-II, ADMETlab).

Response5:

We sincerely thank the reviewer for this suggestion. As requested, we have now cited the official reference articles for all the key databases and tools used in this study, including SwissTargetPrediction, STRING, ProTox-II, and ADMETlab. These citations have been added in lines 84–91 of the revised manuscript and are highlighted in yellow for your convenience.

Comments6: Ensure that reference formatting follows PLOS ONE style (numbered citations in square brackets).

Response5:

We thank the reviewer for highlighting this requirement. We have now reformatted all references throughout the manuscript using the "PLOS ONE" citation style in Zotero to ensure full compliance with the journal's formatting guidelines. Should any references still not fully meet the journal's requirements, we would be grateful for specific indications and will promptly make the necessary corrections.

---

## [Decision Letter · Decision Letter 1]

16 Dec 2025

Dear Dr. Geng,

Thank you for submitting your manuscript to PLOS ONE. After careful consideration, we feel that it has merit but does not fully meet PLOS ONE’s publication criteria as it currently stands. Therefore, we invite you to submit a revised version of the manuscript that addresses the points raised during the review process.

We look forward to receiving your revised manuscript.

Kind regards,

Fazul Nabi

Academic Editor

PLOS One

**Journal Requirements:**

Reviewers' comments:

Reviewer's Responses to Questions

**Comments to the Author**

Reviewer #1: All comments have been addressed

Reviewer #2: All comments have been addressed

2. Is the manuscript technically sound, and do the data support the conclusions?

Reviewer #1: Yes

Reviewer #2: Yes

3. Has the statistical analysis been performed appropriately and rigorously?

Reviewer #1: Yes

Reviewer #2: Yes

4. Have the authors made all data underlying the findings in their manuscript fully available?

Reviewer #1: Yes

Reviewer #2: Yes

5. Is the manuscript presented in an intelligible fashion and written in standard English?

Reviewer #1: Yes

Reviewer #2: Yes

Reviewer #1: (No Response)

Reviewer #2: Reviewer Comments – Second Round

I would like to thank the authors for the extensive work carried out in preparing this revised version. The manuscript shows clear improvements in clarity, structure, and methodological detail. I appreciate the thoughtful responses to the previous comments and the effort invested in strengthening the study. Only a few minor points remain that, once clarified, will further enhance the transparency and interpretability of the manuscript.

1. Molecular docking methodology

The additional information provided in the supplementary material is very helpful. To improve reproducibility directly from the main text, it would be beneficial to briefly include: the exhaustiveness parameter used in AutoDock Vina, the number of poses generated, and whether any positive or negative controls were considered (or a short explanation if they were not used). A concise addition to the Methods would be sufficient.

2. Interpretation of in silico findings

The authors have made an effort to adjust several statements in the Discussion, which is appreciated. For clarity, I kindly suggest slightly softening a few remaining phrases that may appear to imply causality based solely on in silico predictions. Emphasizing the exploratory and hypothesis-generating nature of network toxicology and docking will help align the interpretation with the strengths of these methods.

3. Enrichment analysis presentation

The enrichment analysis is now clearer with the inclusion of FDR correction. If possible, adding metrics such as geneRatio or a similar indicator in the supplementary tables would give readers additional context regarding the weight of each enriched category.

Overall assessment

The manuscript is very close to final form. The remaining points are minor and should be straightforward to address. I appreciate the authors’ careful revisions and believe the study will be strengthened with these small clarifications.

Recommendation: Minor Revision

**Do you want your identity to be public for this peer review?** For information about this choice, including consent withdrawal, please see our Privacy Policy

Reviewer #1: **Yes:**  Prof. Dr. Syed Zahid Ali Shah

Reviewer #2: No

---

## [Author Response · Author response to Decision Letter 2]

17 Dec 2025

Comments1: Molecular docking methodology

The additional information provided in the supplementary material is very helpful. To improve reproducibility directly from the main text, it would be beneficial to briefly include: the exhaustiveness parameter used in AutoDock Vina, the number of poses generated, and whether any positive or negative controls were considered (or a short explanation if they were not used). A concise addition to the Methods would be sufficient.

Response1:

We sincerely thank the reviewer for this excellent and practical suggestion to enhance methodological clarity. We have now added the requested details to the "Molecular Docking" section (Section 2.4) as follows:

“Molecular docking simulations were performed using AutoDock Vina 1.2.2 with the exhaustiveness parameter set to 32, generating nine binding poses for each ligand–target pair. The binding conformation with the lowest predicted binding energy was selected for subsequent analysis. No positive or negative control compounds were included in this study, as the primary aim was to comparatively evaluate the binding affinities and interaction patterns of the candidate drugs with their respective targets under identical docking conditions”.

This addition is found in lines 122-128 of the revised manuscript. We believe this makes our docking protocol fully transparent and reproducible. We are grateful for this suggestion and are happy to make further adjustments if needed.

2. Interpretation of in silico findings

The authors have made an effort to adjust several statements in the Discussion, which is appreciated. For clarity, I kindly suggest slightly softening a few remaining phrases that may appear to imply causality based solely on in silico predictions. Emphasizing the exploratory and hypothesis-generating nature of network toxicology and docking will help align the interpretation with the strengths of these methods.

Response2:

We are very grateful to the reviewer for this critical observation and guidance. We fully agree that the language must accurately reflect the predictive nature of our computational analyses.

We have therefore carefully reviewed the Discussion and Conclusion sections to soften statements that could be misinterpreted as implying direct causality. Key changes include replacing definitive terms with more tentative language to frame our findings as exploratory. These revisions are highlighted in yellow in lines 192-194, 209-211, 214, 239-240, 258, 326, 328, and 337-338.

We thank the reviewer for helping us improve the precision of our interpretation.

3. Enrichment analysis presentation

The enrichment analysis is now clearer with the inclusion of FDR correction. If possible, adding metrics such as geneRatio or a similar indicator in the supplementary tables would give readers additional context regarding the weight of each enriched category.

Response3:

We thank the reviewer for this helpful suggestion. Accordingly, we have added the GeneRatio metric to the enrichment analysis results. The updated supplementary tables containing this information have been uploaded with the resubmission. We appreciate this comment and are happy to make further changes if necessary.

---

## [Editor Report · Decision Letter 2]

4 Jan 2026

Potential mechanisms and effects of AFB1-induced asthma: a comprehensive analysis based on network toxicology and molecular docking

PONE-D-25-52273R2

We’re pleased to inform you that your manuscript has been judged scientifically suitable for publication and will be formally accepted for publication once it meets all outstanding technical requirements.

Kind regards,

Fazul Nabi

Academic Editor

PLOS One

---

## [Editor Report · Acceptance letter]

7 Jan 2026

PONE-D-25-52273R2

PLOS One

Dear Dr. Geng,

I'm pleased to inform you that your manuscript has been deemed suitable for publication in PLOS One. Congratulations! Your manuscript is now being handed over to our production team.

Kind regards,

on behalf of

Dr. Fazul Nabi

Academic Editor

PLOS One